# Shedding light on dark figures: Steps towards a methodology for estimating actual numbers of COVID-19 infections in Germany based on Google Trends

**Christina H. Maaß**[ID]*

Department of Economics, University of Hamburg, Hamburg, Germany

* christina.maass@uni-hamburg.de

## Abstract

In order to shed light on unmeasurable real-world phenomena, we investigate exemplarily the actual number of COVID-19 infections in Germany based on big data. The true occurrence of infections is not visible, since not every infected person is tested. This paper demonstrates that coronavirus-related search queries issued on Google can depict true infection levels appropriately. We find significant correlation between search volume and national as well as federal COVID-19 cases as reported by RKI. Additionally, we discover indications that the queries are indeed causal for infection levels. Finally, this approach can replicate varying dark figures throughout different periods of the pandemic and enables early insights into the true spread of future virus outbreaks. This is of high relevance for society in order to assess and understand the current situation during virus outbreaks and for decision-makers to take adequate and justifiable health measures.

## Introduction

There are many economic phenomena that cannot be measured, such as the true extent of illegal migration, drug trafficking, tax evasion or the spread of diseases. Past research has tackled these dark figures by quantitatively estimating the size of a shadow economy (see e.g. [1]). There, the relationships between the shadow economy and its causes and indicators are calculated with a so-called Multiple Indicators Multiple Causes model and defined with an index of the same name [1]. These proxies were helpful for a better comprehension of hidden markets. Yet with the availability of new data sources that come with the data economy, new opportunities are emerging. High-frequency data from social media, search engines, digitized news and information, allows the application of new techniques to shed light on so far unmeasurable activities.

This is where this paper comes in. We develop a methodological approach to investigate a dominant topic from the beginning of the 2020s – the coronavirus pandemic, and in particular the dark figure of coronavirus infections in Germany based on information obtained from Google Trends. Our approach is novel in that it applies big data analysis, in particular Google

**Funding:** The author received no specific funding for this work.

**Competing interests:** The authors have declared that no competing interests exist.

Trends search volume analysis, to the specific setting of COVID-19 in Germany and approximates the true number of virus infections (including the dark figure) based on this search volume. To the best of our knowledge, no other study has used this data source to investigate unreported cases of COVID-19. Since the outbreak of severe acute respiratory syndrome coronavirus type 2 (SARS-CoV-2) (a beta coronavirus identified in early 2020 as the causative agent of COVID-19 [2]) that causes the highly contagious coronavirus disease 2019 (COVID-19) in late 2019, governments and individuals worldwide are attempting to assess the severity of the coronavirus pandemic. For this purpose, they must rely on the number of new confirmed COVID-19 cases as published by health departments. Considering this data more profoundly it appears that the statistic tools used are outdated and problematic. Since not everyone infected with SARS-CoV-2 is automatically tested and potentially registered, the true number of infections has to be much higher than official statistics. Additionally, people that are tested do not constitute a random sample that could be extrapolated to the whole population, because the probability to be tested is biased towards symptomatic individuals, so that asymptomatic individuals are underrepresented [3]. Given these difficulties, there is great interest in new methodological approaches to obtain reliable estimates of the actual number of infections; it should be borne in mind that most realistic infection quantities are required to understand the dynamics of the pandemic as health measures taken need to be adequate and justifiable.

As a benchmark to get a good picture of the actual number of infections in a country or region, a random sample would be helpful, which, unfortunately, has not been collected. What comes closest are mass tests. However, the positivity rate in such settings appears rather low with 1% in Slovakia [4] and between 0.1% and 0.5% in Austria [5], indicating, for example, about 14% more infections in Lower Austria than reported when extrapolating this result to the total population there. These findings suggest a distorted selection process. Most likely, only individuals with certain characteristics (e.g. asymptomatic and willing to receive a result, or working part-time and thus, at the same time having a lower risk of an infection) took the opportunity to test.

In the analysis of the first German super-spreading event in the municipality Heinsberg, 15.5% of the individuals in the random sample were infected with the coronavirus, which is five times the reported number of infections [6]. However, due to the carnival activities, the situation there was very specific and cannot be easily transferred to other municipalities.

Scientific approaches based on mathematical models have also been applied to estimate the actual number of infections. Liu [7] use an epidemic model based on the law of mass action to estimate reported and unreported COVID-19 cases. They apply an exponential best-fit method in order to estimate the model parameters and reveal that 70% of the infections in Germany and 90% in South Korea are reported, while in the UK, the proportion amounts to 10%. Italy, France and China exhibit quotas between 40% and 60%. Wu [8] apply a semi-bayesian probabilistic analysis in order to account for imperfect testing and imperfect diagnostic accuracy. The model gives a 3 to 20 times higher number of infections than the number of confirmed cases in the United States. 86% of this difference appears to be due to incomplete testing, 14% due to imperfect test accuracy [8]. Using a mathematical nowcast model, Gu [9] defines the true prevalence of corona infections in each US state with an adjusted positivity rate and the official number of confirmed cases. In January 2021, he estimates the true number of infections in the USA to be 2 to 4 times higher than the number of reported cases, which corresponds to a 25–50% detection rate [9]. Correcting data on the test positivity rate in four Canadian cities, the prevalence of infection is estimated to be around twelve times higher than reported [10].

Overall, current research reveals a wide range of possible true infection rates, with the actual number of infections estimated to be several percent up to 20 times higher than the number of reported cases. The goal of this paper is to shed further light into the true figure of SARS-CoV-2 infections in Germany based on big data and in particular search engine data. The theoretical basis for this econometric paper is standard microeconomic theory, i.e. individuals maximizing their utility with the possibility to obtain disease-related information anonymously. We expect this utility to be higher for individuals with symptoms then for purely interested persons. By estimating the volume of COVID-19-related online searches with autoregressive models with and without information about disease-related news articles, [11] show that Google searches are rather representing trends in infection levels than being mainly driven by outbreak news. Although the search volume data surely imply some error in our analysis, it appears that Google Trends data is a viable and highly insightful option to improve standard statistics (that include errors on their own) in order to deal with the complicated task of monitoring infection trends (see [11–14]).

As an exemplary country, Germany–the largest EU economy and a country with an elaborate health and surveillance system–is investigated. In Germany, the public health institute Robert Koch Institute (RKI) publishes the confirmed COVID-19 numbers for the entire federal territory as well as for the federal states on a daily basis. Our approach aims to correct the number of registered infections for unregistered cases to get closer to the true infection rate. As a proxy for the actual incidence we use the volume of coronavirus related searches (*corona test*, *quarantine*, *test center*, *loss of smell*, *loss of taste*) obtained from Google Trends [15]. Since we expect high correlation between registered and actual infection numbers, we analyze the relationship between search volume and official case numbers. We find substantial and significant correlation between the two during the first three coronavirus waves in 2020 and 2021 at the state and national level. Using a vector autoregressive (VAR) model and applying a Granger Causality Test it is shown that the search volume is causal for the development of registered cases. Finally, based on the level of correlation with reported infections, we approximate the magnitude of true infections and show that underreporting varies across the different phases of the pandemic.

The remainder of the paper is organized as follows. Section 2 gives an overview of specific features of the analysis. Section 3 covers the data and explains the method used. In section 4, we present the results. In Section 5, the results are discussed. Section 6 concludes the paper.

## Features of the analysis

### Unreported cases

The true number of COVID-19 infections includes both reported and unreported cases. Being able to estimate the number of unreported cases, as unknown element of the actual number of infections, is of central interest for fighting a pandemic and for taking adequate action on limiting the virus spread.

The unreported cases can be assigned to different groups depending on the circumstances of their missed registration and whether the infection is asymptomatic or symptomatic (see Fig 1). The asymptomatic group consists of individuals who carry the infection inside, but do not notice it, because they exhibit only slight symptoms or no symptoms at all. They can either be presymptomatic, meaning that they develop symptoms later on, or completely asymptomatic, meaning that they do not develop symptoms at all. Both values seem to be substantial but vary between different studies. The number of asymptomatic infections, that remained asymptomatic, was estimated to be 18% on a cruise ship in Japan [16], 22% in a German village with a super-spreading event [6], 30% in the US in general [17], 31% on charter flights evacuating

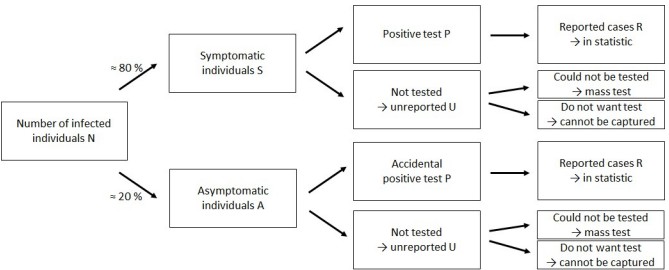

**Fig 1. Composition of true number of infections.**

Japanese citizens from Wuhan [18], and larger than 40% in the municipality of Vo' in Italy [19]. In an overview of more than 40 studies it was revealed that around 16% of infections are asymptomatic [20]. Sah [21] state that estimates on asymptomatic courses often contain a downward bias. When correcting for this, around 35% of infections can be characterized to be truly asymptomatic [21]. Based on this evidence, we rather conservatively estimate the proportion of asymptomatic infections to be 20%. Asymptomatic infections are of high relevance for the spread of COVID-19, since around 40% [6,22] to approximately 50% [17,23] of infections occur in the asymptomatic phase of the infector (before the onset of symptoms) and asymptomatic individuals are found to be at least as infectious [19,24] or at least 75% as infectious as symptomatic individuals [17]. When these individuals are not tested accidentally or because they become symptomatic over time, they are not captured in the official statistics.

Symptomatic individuals, conversely, notice the infection. If test capacities are limited or individuals do not meet the requirements of being a contact of a COVID-19 positive person or having been in a risk region–both of which were especially the case early in the pandemic–they face a similar situation as asymptomatic individuals. They do not know their infection status and may belong to the group of unreported cases. These individuals as well as the asymptomatic individuals who were not accidentally tested can then only be detected by later mass or antibody tests (depending on the type of test, a corona infection is only detectable for a certain period of time after the infection). Additionally, there are infected individuals that do not want to reveal their infection. When health authorities do not notice them, they are not going to be tested and will therefore not appear in the official statistics. They would neither go to public mass tests. Apart from that, there are some wrong negative test results. However, this number is estimated to be very small for PCR tests (below 1%), so that we abstain from this possibility here [25].

With our approach, we are able to detect the unregistered symptomatic infections. Since individuals that consider themselves as healthy are also of high importance, possible dark figure estimates based on this method should be corrected for the percentage of infections that occur asymptomatically in order to gain further insights about the population as a whole.

## Big data and Google Trends

With the data economy, an infinite amount of information is collected every day. In order for scientific research to benefit from these developments, it is imperative to make use of big data sources like social media, search engines and digitized information (for more details on big data analysis see e.g. [26] or [27]). For this reason, this paper combines theoretical considerations not only with data from public authorities and institutions but also with big data. This brings the advantage of high frequency and timeliness, which can be used to filter out the true number of coronavirus infections.

Several studies indicate the suitability of big data for disease monitoring (see e.g. [12]). Results from social media (including search queries) are highly correlated with more traditional surveillance programs. In addition, they provide the advantages of higher effectiveness and fast detection of disease outbreak developments in real-time as well as easy and low-cost access. Disadvantages are the possibility for false positive and false negative results, the difficulty to eliminate background noise as well as the limited representativeness, which can induce biases. Therefore, it is often recommended to use this method rather in addition to existing surveillance approaches [12].

It is important to take into account that the different data sources exhibit different characteristics. When collecting data via Google Trends, through the anonymity of the searches, it is possible to obtain "revealed and not stated users´ preferences" [28], so that we receive unfiltered and unprocessed information [28,29]. In this respect, we expect Google search queries to be particularly suitable to measure intrinsic motivation, so-called implicit motives. They are measured indirectly, whereas posts on social media rather reflect explicit behavior and motives, that are similar to motives usually expressed and measured more directly, e.g. in questionnaires [30]. Applied to our setting, this would mean that individuals that do not wish to reveal their true infection status to others might still issue searches on Google. When they, for example, look up symptoms related to the coronavirus, they are going to be captured by our approach. Compared to surveys, there is less cognitive dissonance with respect to Google searches [31]. However, when substantial emotion is involved or the intention of search queries is unclear, the explanatory power of Google searches can be limited [31].

The possibility of timely information without acquisition cost combined with direct statements of the broad population makes Google Trends data appear highly attractive for analyzing the coronavirus spread. Furthermore, recent research indicates high eligibility of this data source. An analysis of influenza surveillance in Boston shows that Google Trends data is highly insightful for monitoring virus spreads and exhibits substantial predictive power [13]. Current literature on the coronavirus pandemic confirms that Google search queries can be used for COVID-19 surveillance [11] and that the development of search volume on Google can indicate the course of symptoms of the disease [32]. Additionally, there is high correlation between the COVID-19 incidence and Google searches regarding this illness in Colombia [33]. In China, high correlation is observed between confirmed and suspected COVID-19 cases and searches for 'pneumonia' and 'coronavirus' [14].

In general, when working with Google Trends data, one has to keep some pitfalls in mind. Google publishes only relative search volumes, no absolute values, and this volume is obtained from a supposedly representative sample. The data consists of values between 0 and 100 for each period and location and is issued at different frequencies (e.g. daily or weekly) based on the duration of the considered period. A value of 0 means lowest search volume during that period and in that area and a value of 100 means highest volume. Therefore, the selection of the period for which Google Trends data is retrieved has to be selected carefully, since it determines the distribution of the values between 0 and 100. In general, it is recommended to use exactly the same period on Google Trends for which official data is used and analyzed [28]. Furthermore, misspellings, accents, the use of quotation marks and the decision whether to analyze the search term or the topic can influence the result [28]. All of these factors need to be considered when conducting an analysis of Google Trends data.

## Data and method

The number of confirmed COVID-19 cases in Germany is obtained from RKI [34] as publicly available data. At the national level we obtain daily values. For the analysis, the values are

aggregated at the weekly level. The regional cases are published by RKI as 7-days-cases. We take the 7-days-cases number once per week, corresponding to the day of the week to which the searches are aggregated.

The keywords (Google searches) used as a proxy for incidence were chosen by brainstorming in team together with current knowledge on specific COVID-19 symptoms. This gives the keywords *corona test*, *quarantine*, *test center*, *loss of smell*, *loss of taste*, and, in the beginning of the considered period, *pneumonia* (in German: Geruchsverlust, Geschmacksverlust, Testcenter, Quarantäne, Coronatest, Lungenentzündung). They do not include other spellings. However, it can be assumed that the development of related queries will be very similar. Searching for *corona test*, *quarantine* and *test center* is expected to be highly predictive of a coronavirus infection. *Loss of taste* and *smell* correspond to the most frequent yet distinct symptoms of the virus. *Pneumonia* is viewed in order to detect possible early COVID-19 cases before the coronavirus officially reached Germany. The volume for these Google searches in Germany is publicly available via Google Trends [15] and obtained for the period October 2018, one winter before the beginning of the coronavirus pandemic, until May 2021, the end of the third wave and the last month with confinement measures in place after the second lockdown. The data collection and analysis were conducted in compliance with the terms and conditions for the two public data sources. The sources are explicitly referred to in the text and named in the reference list.

In order to demonstrate the suitability of Google search volume to estimate the true number of COVID-19 infections in Germany, we investigate the correlation between search volume related to the COVID-19 pandemic and officially registered coronavirus cases. Initially, the correlation between the Google searches and COVID-19 infections is calculated, first for the whole period and second for the period of the pandemic from 03/2020 until 05/2021 (COVID-19 case numbers are officially available from RKI since February 25[th] 2020). We use the Spearman´s rank order correlation coefficient as elaborated by [35,36] for two reasons. Firstly, since neither the search volume nor the number of infections are normally distributed, we needed an alternative method to the Pearson correlation. In such a case, the mayor recommendation is to use the Spearman correlation due to higher validity of a non-parametric method when analyzing non-parametric data [37]. Secondly, in order to be able to establish a correlation between two series, the values of both should be appropriately comparable [35]. Since this is not the case when analyzing the absolute amount of registered COVID-19 infections together with relative search volume, it is helpful to sacrifice some informational value and to compare the two by rank.

Subsequently, we address the question whether there is only correlation or also a causal relation between search volume and confirmed COVID-19 cases. For this purpose, we set up a VAR model that explains the development of the keyword and RKI variables, respectively, based on the past values of the dependent variable as well as the independent variables thus accounting for their joint past evolution [38]. It was tested for causality using the Granger Causality Test based on Granger [39] and following the procedure by Toda and Yamamoto [40] and Dolado and Lütkepohl [41]. The procedure indicates that lag-selection tests and in particular the Wald statistics underlying the causality test are valid in case of non-stationary and possibly co-integrated time series under certain circumstances in the levels VAR model. Preconditions are that the time series are at most integrated of order two and that the lag order $p$ is chosen to be at least equal to the correct lag order $k$ plus the maximum order of integration $d_{max}$. After the optimal lag-length has been determined, it is therefore necessary to increase the lag order used for the levels VAR by the maximum order of integration of the time series. With this in hand, the usual approach can be implemented to test whether one time series is helpful for forecasting another time series and is thus Granger causal for this series. Especially

in cases where the underlying dataset has relatively few variables and numerous lags–as is the case with our data–the efficiency loss associated with the overfitting should be less of a concern than possible pretest biases when testing for stationarity and cointegration [40].

In the last part of the analysis, we approach the order of magnitude of true infections compared to official RKI numbers. We assume perfect correlation between search volume and actual infection levels and define the dark figure as the gap between obtained and perfect correlation. We calculate the correlation coefficients for different sub-periods to check whether the dark figure varies across the different stages of the pandemic.

For the analysis we use Python 3.9.

## Results

### National level

In order to get an overview of the development of searches for our keywords during and before the pandemic, we graphically examine the search volume since one winter before the onset of the pandemic until May 2021. By including one winter period before, it is ruled out that the surge in search volume during/after the 2019/20 winter season was purely driven by seasonal fluctuations. Mapping the search volume for *pneumonia* together with the COVID-19 cases by RKI results in no discrepancy regarding the beginning of the pandemic. The graph reveals a clear surge in queries in mid-March 2020 matching a substantial increase in COVID-19 cases (see S1 Fig). A few weeks after the appearance of the new coronavirus, queries drop to usual levels. This implies a first correspondence between Google searches related to the coronavirus and the evolution of the virus cases.

The development of Google search queries for the three keywords, *corona test*, *test center* and *quarantine* shows no abnormalities in the period prior to the pandemic (see S2 Fig). The search queries on the two coronavirus symptoms *loss of taste* and *loss of smell* fluctuate already before the beginning of the pandemic, since these symptoms can also be connected to other illnesses (see S3 Fig). During the three virus waves, the search volume for all terms clearly rises, mimicking the development of infection levels (see Fig 2). During the first coronavirus wave (03–05/2021) interest in the term *quarantine* is the highest, in the second wave (10/2020–01/2021) *loss of smell* and *loss of taste* show the highest volume, and in the third wave the amplitude is highest for *corona test* and *test center*. The search volumes are all obtained separately, prohibiting any inference on the ratio of queries between the different keywords.

Comparing the search volumes with the development of COVID-19 cases it becomes clear that the term *pneumonia* is able to indicate the onset of the coronavirus pandemic and the selected keywords for the analysis are able to depict the first three coronavirus waves in Germany. Additionally, the average search volume for the five keywords and the registered COVID-19 cases, both normalized to values between 0 and 100, evolve quite closely (see Fig 3). This constitutes a good starting point for checking possible correlations between the search queries and official COVID-19 cases.

For this purpose, we calculate the Spearman rank-order correlation coefficient for each search term except pneumonia (the correlation between reported infections and pneumonia is negative, since with the growing amount of COVID-19 infections the term COVID-19 (coronavirus) was used instead of pneumonia). We thereby distinguish between two different periods: the whole period from 10/2018 (one winter season before the outbreak of COVID-19) until 05/2021 as well as the pandemic period, starting with the month in which the coronavirus had arrived completely in Germany (03/2020) until the end of the third wave (the beginning of summer) in May 2021. The values displayed in the following tables are significant at the 5% level.

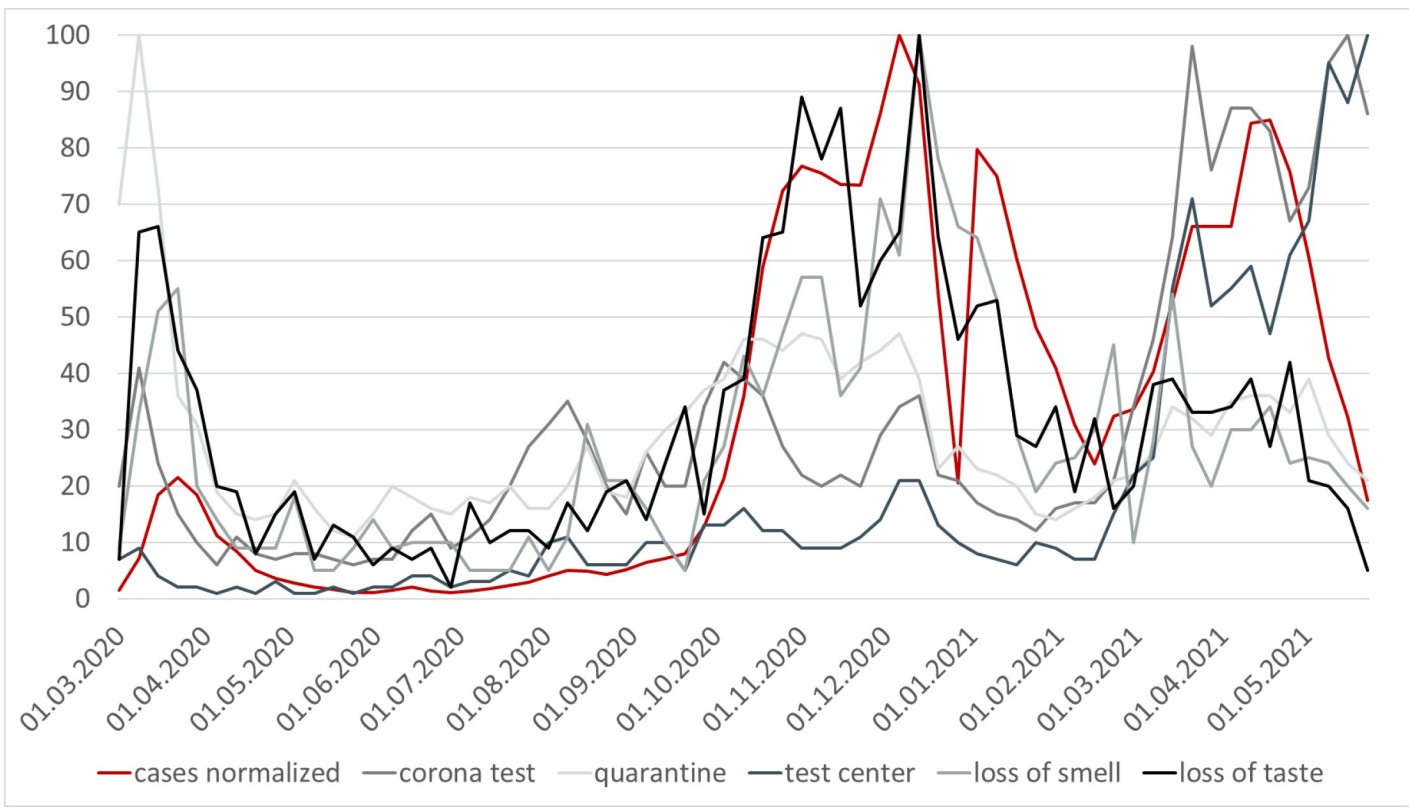

**Fig 2. Development of normalized COVID-19 cases and search volume (03/2020–05/2021).** Note: Cases normalized to a value between 0 and 100 to achieve better comparability with the search volume.

During the period 10/2018 until 05/2021 (139 weeks), the correlation of reported infections is highest for the search term *corona test* (0.95), followed by *quarantine* (0.90) as can be seen in the first row in Table 1. Slightly lower correlations exhibit the keywords and *loss of smell* and *loss of taste* with 0.87 as well as *test center* with 0.86. During the period of the pandemic, the correlation between the search queries and the official cases is substantially lower for the terms *corona test* and *quarantine*, lower for *test center* as well as *loss of smell* and only slightly lower for *loss of taste* (see second row of Table 1). The correlations for this specification range from 0.59 for *corona test* and *quarantine* to 0.81 for *loss of taste*. This indicates suitable keyword selection and a substantial correlation between keywords and registered infections.

In order to get a first impression of the possible nowcasting character of the dataset, the search queries are correlated with the RKI numbers of the subsequent week (lag 1). For a nowcast of COVID-19 incidence based on fitness tracker data see [42]. This reduces the correlation coefficient for all specifications (see S1 Table). This finding points to delays between the official numbers and the issued queries of less than one week.

## Federal state level

In order to validate the suitability of Google searches for estimating the actual number of COVID-19 cases, we look at the correlation from a regional point of view and collect the Google query volumes for each federal state separately. During this exercise, it becomes apparent that the data quality for the regional searches is rather weak and not sufficient for in-depth analyses. Especially in the least populous states Bremen, Mecklenburg-Western Pomerania,

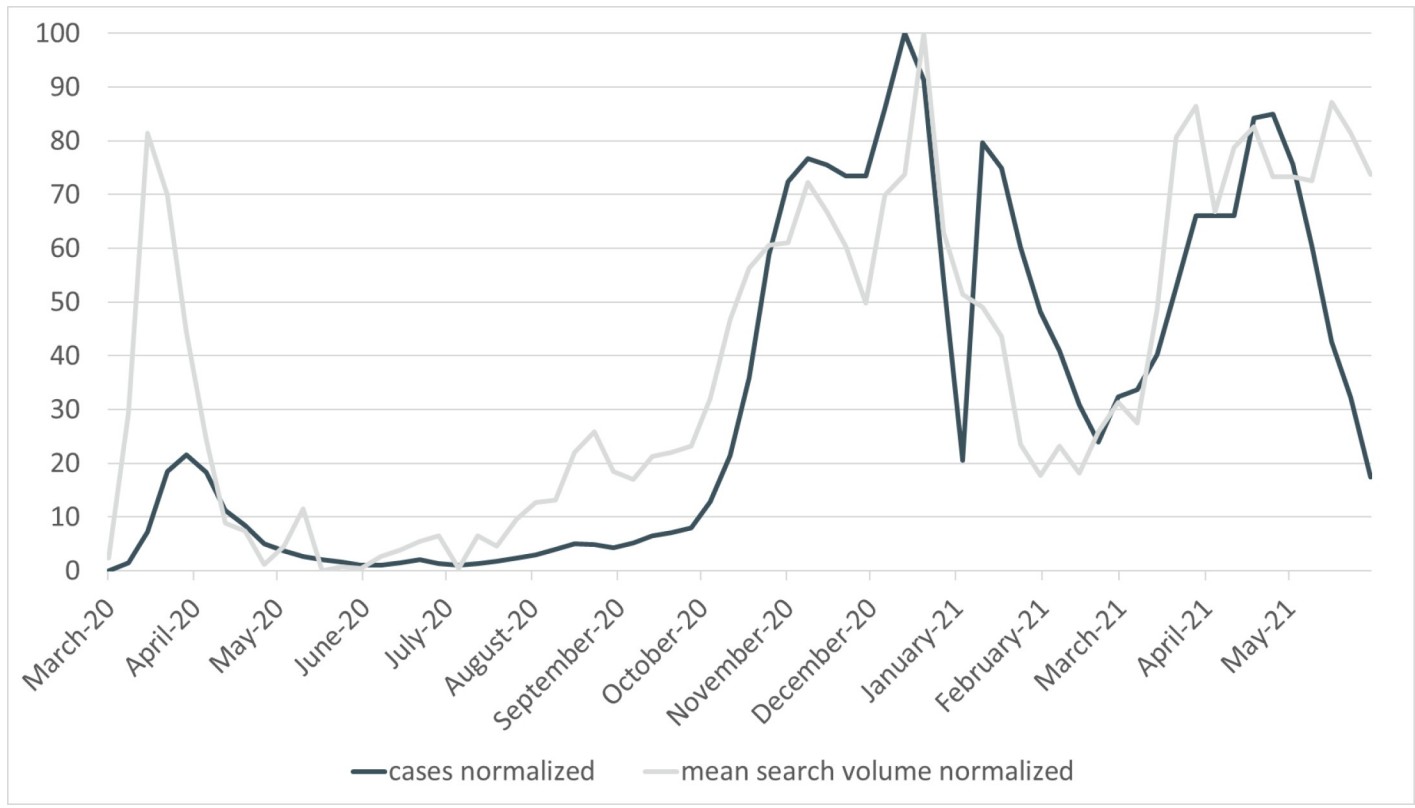

**Fig 3. Development of normalized COVID-19 cases and normalized mean search volume (03/2020–05/2021).** Note: Cases normalized to a value between 0 and 100 to achieve better comparability with the search volume. The mean search volume is calculated for the five keywords and then normalized again to a value between 0 and 100.

and Saarland [43], the total search volume is often too low to obtain (meaningful) values from Google Trends. Due to missing data points and frequent zero entries, establishing correlations for these states is not informative and they are dropped from the subsequent analysis.

For the remaining states, the correlation between the search volumes and the countrywide case numbers by RKI (as used in the countrywide analysis) is displayed in Table 2. Substantial differences stand out between the federal states. For example, in Thuringia the search term loss of taste gives a correlation of 30% and test center gives 25%, whereas the correlation in Bavaria is 68% for loss of taste and 73% for test center. 61 out of the 65 correlation coefficients are significant.

In a next step, we check the correlation between search volumes per state and case numbers per state (see S2 Table). We find that the correlation decreases in most states with respect to the term corona test. There is a tendency for lower correlation for test center. Furthermore, the pattern indicates higher correlation for quarantine and a tendency to higher correlation

**Table 1. Correlation of search queries with COVID-19 cases whole period and pandemic period.**

| period | corona test | quarantine | test center | loss of smell | loss of taste |
|---|---|---|---|---|---|
| 10/2018–05/2021 | 0.9491 | 0.9028 | 0.8559 | 0.8740 | 0.8686 |
| 03/2020–05/2021 | 0.5902 | 0.5894 | 0.6978 | 0.7862 | 0.8123 |

Note: Table depicts the Spearman rank-order correlation coefficients. All values are significant at the 5% level.

**Table 2. Correlation of national case numbers with Google search queries per state (03/2020–05/2021).**

| federal states | corona test | quarantine | test center | loss of smell | loss of taste |
|---|---|---|---|---|---|
| Baden-Wuerttemberg | 0.4793 | 0.5978 | 0.6236 | 0.3327 | 0.5369 |
| Bavaria | 0.6318 | 0.5720 | 0.7283 | 0.5411 | 0.6815 |
| Berlin | 0.5454 | 0.4627 | 0.4923 | 0.3647 | 0.3451 |
| Brandenburg | 0.5828 | 0.5521 | not significant | 0.3348 | 0.4006 |
| Hamburg | 0.4964 | 0.4350 | 0.3773 | 0.3209 | 0.3038 |
| Hesse | 0.5993 | 0.5728 | 0.6723 | 0.5005 | 0.5047 |
| Lower Saxony | 0.5444 | 0.5141 | 0.3889 | 0.2814 | 0.3192 |
| North-Rhine-Westphalia | 0.4438 | 0.4624 | 0.6347 | 0.6493 | 0.5178 |
| Rhineland-Palatinate | 0.4336 | 0.4970 | 0.4242 | 0.4026 | 0.3378 |
| Saxony | 0.6098 | 0.6718 | 0.5773 | 0.5169 | 0.4082 |
| Saxony-Anhalt | 0.5442 | 0.6194 | 0.3976 | 0.3808 | not significant |
| Schleswig-Holstein | 0.6636 | 0.3204 | 0.4370 | not significant | 0.3395 |
| Thuringia | 0.4851 | 0.6295 | 0.2477 | not significant | 0.2987 |

Note: Table depicts the Spearman rank-order correlation coefficients. All values are significant at the 5% level.

for loss of smell. For loss of taste, the direction of the change in correlation is not clear. The lowest correlation refers to test center in Brandenburg (0.26), the highest correlation stems from quarantine in Saxony (0.79).

It becomes clear that the correlations at the federal state level are substantially lower than at the national level. This might be due to the fact that regional blur is corrected for and regional differences are averaged out in the Germany-wide analysis. Additionally to the few search queries, there might be different delays until the official cases are published in each state, which makes the correlations lower when looking at the states separately. Different population densities or the number of inhabitants might partly drive the large differences between the federal states, as a certain search volume is necessary to derive useful results. In general, however, it is questionable how precisely individuals are assigned to the states, since people spend a lot of time in states in which they do not live due to work, school or because of second homes and commuting. Overall, the search volume per state is substantially lower, contains significantly more zero entries and is overall less representative than the country-wide search queries. Still, the considerable correlations confirm the link between Google search queries related to the coronavirus and reported coronavirus cases, although the exact correlations should be interpreted with care.

## Testing for a causal link between Google searches and COVID-19 infections

Given the promising results of the previous sections, we now move one step further and test for a causal link between the Google search time series and the time series of reported COVID-19 infections during the pandemic period. For this purpose, following the procedure introduced by Toda and Yamamoto [40], we construct a VAR model and perform the Granger causality test for all variables in both directions. We start by checking whether the time series are stationary using the Augmented Dickey-Fuller (ADF) and Phillips-Perron unit root test as well as the Kwiatkowski-Phillips-Schmidt-Shin (KPSS) test. While the Phillips-Perron and KPSS test point at some series being stationary and towards a maximum order of integration equal to 1, the ADF test indicates that all keywords are integrated of order 1, except *test center*, which is integrated of order 2 (see S3 Table). The maximum order of integration $m$ thus is

equal to 2. In the next step, we set up a standard VAR model in the levels of the data. In order to determine the optimal lag length, we use the Akaike Information Criterion (AIC), Schwarz-Bayesian Information Criterion (BIC) and Hannan-Quin Information Criterion (HQIC). These tests point to an optimal lag length p = 12 (see S3 Table). Thirdly, we test for cointegration between the different time series using the Johansen test and find a cointegrating rank of 3 (see S5 Table). Cointegration with official RKI case numbers only exists for the search term *loss of smell*. Since Toda and Yamamoto [40] suggest to overfit the VAR model through increasing the optimal lag order by the maximum order of integration, the VAR model is fitted with p = 14.

$$
\begin{pmatrix} RKI_t \\ corona\ test_t \\ quarantine_t \\ test\ center_t \\ loss\ of\ smell \\ loss\ of\ taste \end{pmatrix} = \begin{pmatrix} m_1 \\ m_2 \\ m_3 \\ m_4 \\ m_5 \\ m_6 \end{pmatrix} + \begin{pmatrix} a^1_{1,1} & \cdots & a^1_{1,6} \\ \vdots & \ddots & \vdots \\ a^1_{6,1} & \cdots & a^1_{6,6} \end{pmatrix} \begin{pmatrix} RKI_{t-1} \\ corona\ test_{t-1} \\ quarantine_{t-1} \\ test\ center_{t-1} \\ loss\ of\ smell \\ loss\ of\ taste_{t-1} \end{pmatrix} + [\ldots] + \begin{pmatrix} a^{14}_{1,1} & \cdots & a^{14}_{1,6} \\ \vdots & \ddots & \vdots \\ a^{14}_{6,1} & \cdots & a^{14}_{6,6} \end{pmatrix} \begin{pmatrix} RKI_{t-14} \\ corona\ test_{t-14} \\ quarantine_{t-14} \\ test\ center_{t-14} \\ loss\ of\ smell_{t-14} \\ loss\ of\ taste_{t-14} \end{pmatrix} + \begin{pmatrix} u_{1t} \\ u_{2t} \\ u_{3t} \\ u_{4t} \\ u_{5t} \\ u_{6t} \end{pmatrix}
$$

In order to make sure the model is well specified, it is tested for serial correlation in the residuals using the Durbin Watson test. With this specification, there is no indication for serial correlation (see S6 Table). Finally, we are ready to perform the Granger causality test. For this test, we apply the optimal lag order determined earlier. The p-values for the null hypothesis of no Granger causality can be found in Table 3. It is read as indicating whether one column variable (predictor) Granger causes one row variable (response).

It turns out that the null hypothesis of no Granger causality for RKI cases can be rejected at the 10% level for all search terms, at the 5% level for all terms but *corona test* and at the 1% level for all variables except *corona test* and *test center*. On the other hand, the null hypothesis that the number of registered infections is not Granger causal for the development in search volume can be rejected at the 1% level for *loss of smell* and *loss of taste* and at the 5% level for *quarantine*. This indicates a partially reciprocal relationship between search queries and registered infections. More importantly, however, the development in search volume is found to be predictive of current infection levels. The result is robust to the inclusion of a time trend that controls for the different months of a year. The findings support the thesis that the volume of searches issued on Google can be used for estimates of the true figure of COVID-19 infections. Normally, a possible cross check can be performed through verifying that cointegrated variables are indeed causal. Since only *loss of smell* is cointegrated with RKI, it is only possible to do the cross check for this combination. It turns out that we do not have conflicting results, since *loss of smell* is also Granger causal for RKI and vice versa.

**Table 3. Results Granger causality test.**

| y/x | RKI | corona test | quarantine | test center | loss of smell | loss of taste |
|---|---|---|---|---|---|---|
| RKI | 1.0000 | **0.0772** | **0.0000** | **0.0319** | **0.0000** | **0.0000** |
| corona test | 0.6705 | 1.0000 | 0.0304 | 0.0000 | 0.0000 | 0.0159 |
| quarantine | 0.0430 | 0.1148 | 1.0000 | 0.3584 | 0.0588 | 0.0303 |
| test center | 0.1440 | 0.0000 | 0.0413 | 1.0000 | 0.0000 | 0.0011 |
| loss of smell | 0.0000 | 0.1332 | 0.0002 | 0.5962 | 1.0000 | 0.0000 |
| loss of taste | 0.0004 | 0.0046 | 0.0000 | 0.0142 | 0.0000 | 1.0000 |

Note: Table indicates p-values from testing for Granger non-causality of column variables for row variables.

**Table 4. Correlation in different sections of the pandemic.**

| period | corona test | quarantine | test center | loss of smell | loss of taste | average | dark figure |
|---|---|---|---|---|---|---|---|
| 03/2020–05/2020 | 0.5376 | not significant | not significant | 0.6661 | 0.8524 | 0.6854 | 31% |
| 06/2020–10/2020 | 0.8305 | 0.8689 | 0.9050 | 0.5800 | 0.8440 | 0.8057 | 19% |
| 11/2020–01/2021 | 0.5437 | 0.5740 | not significant | not significant | 0.5881 | 0.5686 | 43% |
| 02/2021 - 05/2021 | not significant | 0.8054 | not significant | not significant | 0.6429 | 0.7242 | 28% |

Note: Second from left part in the table depicts the Spearman rank-order correlation coefficients. All values are significant at the 5% level.

## Subdivision of the time series in different sections and derivation of a dark figure estimate

In order to understand the connection between search queries and COVID-19 cases more profoundly, the time series is split into different sections of the pandemic. The first section (03-05/2020) refers to the first virus wave, the second section (06-10/2020) refers to the calmer summer period, the third section refers to the period around Christmas (11/2020–01/2021) and the fourth section refers to the third wave (02-05/2021). This exercise aims at detecting different levels of the dark figure at different points in time during the pandemic.

In the first wave, the test capacities were quite limited and infections were expected to be mainly relevant for individuals with contact to China or Chinese citizens. Both facts should lead to a rather substantial amount of unregistered infections. During the summer period, the occurrence of infection was much lower as should be the dark figure. In the second wave, we again expect a high level of undetected COVID-19 infections since there was substantial occurrence of infection. Especially during the Christmas holidays on the one hand, the possibilities to be tested were much lower, and on the other hand, many infections were reported with a considerable delay. During the last section, we expect the number of unregistered infections to decline, since from March 2021 broad testing possibilities were launched with large test centers and self-tests that could be bought for domestic use.

For each of the respective sections, we calculate a rough dark figure, which brings us closer to the true number of infections. For this purpose, we firstly assess the average correlation out of the significant coefficients for that period. It is then assumed that the search volume would be perfectly correlated to the actual occurrence of infections. The dark figure would therefore correspond to the gap between obtained correlation and perfect correlation (= 1). In our specification, it amounts to 31% in the first wave. It is much lower during summer with around 19%. In winter, it is estimated to be highest (43%), and it is substantially reduced again in the last period (28%). Table 4 gives an overview of the values for the four periods. Interestingly, three out of the five coefficients are not significant in the last period. This indicates the advantage of Google searches to detect infected individuals that are not tested, although they would like to know their true infection status, is ending, since everyone has the possibility to be tested from the second month of the considered last period.

## Discussion

We are fully aware that there is some noise (in an econometric sense) in the data set such as some people searching for symptoms or keywords related to the coronavirus may not be infected. Furthermore, not everyone uses Google, but other search engines (although this share is negligible), and some people, especially elderly, do not use search engines at all, but are still partly represented by younger people searching on their behalf. Overall, we expect this

background noise to be rather constant and not to bias our estimates as long as there are no large and significant differences between the age cohorts with respect to both infections and search activity. This does not mean that these factors do not contain valuable information, but that this variation cannot be explained and enters the error term. The achievement of this paper is to infer information from a big data source even though many relevant factors are unknown. Especially, since we do not look at absolute but relative search volume, we believe we get valuable insights into the infection situation. Furthermore, the keywords relevant for COVID-19 incidence may change over time. This is partially visible in the volume plots. Possible remedies would include keyword selection based on machine learning.

For a better comparison between search behavior and registered infections, it would be advantageous to have absolute search volume instead of relative volume. With this kind of data, it would also be possible to obtain dark figures exceeding 100%. In general, our approach likely underestimates the true level of infections due to the incomplete reference of reported infections. Since, as indicated above, this method needs to be employed in addition to official data, we automatically incorporate part of the error of the original data. Especially in the beginning of the pandemic, when the illness is not yet known and test capacities are limited, the reference value (reported infections) is lower. Adding a specific percentage to this number leads to a lower dark figure (as seen in S4 Fig). In the course of the pandemic, when the health system has adjusted, we approach a more realistic assessment of the multiplier. In the end of the pandemic, the informative value decreases again as some kind of self-testing possibilities are available. So, the main conclusion is that there is a dark figure and that it is substantial. In this context, the results from our analysis point to a minimum.

With the described approach, we expect to capture those individuals with big data that have mild symptoms that cannot be tested although they are symptomatic or that do not wish to be detected and would try to avoid public tests. These persons might issue Google searches on COVID-19 related topics. People that do not have any symptoms cannot be captured directly by this approach. For this reason, our dark figure appears rather low. However, it is possible to extend our approach and to correct the calculated figures for the estimated percentage of asymptomatic cases. The easiest way to account for that is to add the approximated percentage of asymptomatic infections from the literature–a conservative approach would be around 20%–to the corrected figure of symptomatic infections, which is around 1.19 to 1.43 times the number of reported infections. This gives a multiplier between 1.43 and 1.72, meaning that adding around 43% to 72% to the number of reported infections by RKI, depending on the phase of the pandemic, brings us closer to the true number of infections. Comparing this approximation to current findings, our result appears quite plausible and confirms substantial underestimation of infections.

## Conclusion

Since current methods could not prove to be sufficient, this paper uses a novel approach based on big data to shed light on the true number of coronavirus infections in Germany. We find substantial correlation between COVID-19 related Google search volumes derived from Google Trends and RKI-confirmed COVID-19 cases in Germany at both the national and regional level. The relationship between search queries and registered COVID-19 cases is shown to be causal by Granger causality tests. By dividing the time series in different sections, we are able to show that the level of correlation and accordingly the approximated dark figure varies through the different phases of the pandemic.

The preceding analysis provides insight into three relevant aspects. First, the extent of the dark figure of COVID-19 infections is relevant, since it is estimated to range between 19% and

43% when ignoring asymptomatic cases and between 43% and 72% when accounting for them. Second, it is pointed out that the dark figure depends on certain variables, namely the search volume for different COVID-19 related keywords. Third, the analysis proves to be helpful for early detection of increased unregistered occurrence of infection in future pandemics.

This paper encourages further in-depth analyses of Google Trends data to make more invisible circumstances visible. Further research also using information from other big data sources like social media or newspaper articles is necessary to better understand the opportunities accompanying the new data economy and to shed light on hidden numbers.

## Supporting information

**S1 Fig. Search queries for pneumonia vs. confirmed cases (10/2018–05/2021).** Note: Logarithm of search queries and cases used.
(TIF)

**S2 Fig. Search volume over time corona test, quarantine, test center (10/2018–05/2021).** Note: The figures do not represent the ratio of search queries to one another, but the distribution of each search query separately over time.
(TIF)

**S3 Fig. Search volume over time loss of smell, loss of taste (10/2018–05/2021).** Note: The figures do not represent the ratio of search queries to one another, but the distribution of each search query separately over time.
(TIF)

**S4 Fig. Cases per week as reported by RKI and adjusted by our multiplier.** Note: Multiplier based on table, corrected for 20% asymptomatic cases.
(TIF)

**S1 Table. Correlation of search queries with COVID-19 cases whole period and pandemic period with one lag.** Note: Table depicts the Spearman rank-order correlation coefficients. All values are significant at the 5% level.
(TIF)

**S2 Table. Correlation of cases per state with search queries per state (05/2020–05/2021).** Note: Table depicts the Spearman rank-order correlation coefficients. All values are significant at the 5% level. As case numbers per state are only available from May 10, 2020, we restrict our analysis to the period from that date until the end of May 2021.
(TIF)

**S3 Table. ADF and Phillips-Perron unit root tests and KPSS test for stationarity.** Note: ADF: Augmented Dickey-Fuller, KPSS: Kwiatkowski-Phillips-Schmidt-Shin. Null hypothesis for ADF and Phillips-Perron: presence of a unit root (time series not stationary); for KPSS test it is stationarity. All tests are performed with the default of only adding a constant.
(TIF)

**S4 Table. Lag selection according to AIC, BIC and HQIC.** Note: AIC: Akaike Information Criterion, BIC: Schwarz-Bayesian Information Criterion, HQIC: Hannan-Quin Information Criterion.
(TIF)

**S5 Table. Johansen trace and maximum eigenvalue tests for cointregration.** Note: Table shows null hypothesis and alternative, the test statistics and 5% critical values for 12 lags

for Johansen trace and maximum eigenvalue test for cointegration.
(TIF)

**S6 Table. Durbin Watson test statistics.** Note: Table depicts Durbin Watson test statistics for p = 14 lags.
(TIF)

## Acknowledgments

The author would like to thank the team around Professor Straubhaar at University of Hamburg for meaningful feedback as well as Paul Griensteidl, M.Sc., and Ronald G. Maaß, MBA, for endless discussions about the interpretation of results.

## Author Contributions

**Conceptualization:** Christina H. Maaß.

**Data curation:** Christina H. Maaß.

**Formal analysis:** Christina H. Maaß.

**Investigation:** Christina H. Maaß.

**Methodology:** Christina H. Maaß.

**Project administration:** Christina H. Maaß.

**Resources:** Christina H. Maaß.

**Software:** Christina H. Maaß.

**Supervision:** Christina H. Maaß.

**Validation:** Christina H. Maaß.

**Visualization:** Christina H. Maaß.

**Writing – original draft:** Christina H. Maaß.

**Writing – review & editing:** Christina H. Maaß.

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
