## [Decision Letter · Decision Letter 0]

15 Aug 2022

PONE-D-22-02453Shedding light on dark figures: Steps towards a methodology for estimating actual numbers of COVID-19 infections in Germany based on Google TrendsPLOS ONE

Dear Dr.Christina Heike Maaß ,

Thank you for submitting your manuscript to PLOS ONE. After careful consideration, we feel that it has merit but does not fully meet PLOS ONE’s publication criteria as it currently stands. Therefore, we invite you to submit a revised version of the manuscript that addresses the points raised during the review process.

We look forward to receiving your revised manuscript.

Kind regards,

Ricky Chee Jiun Chia

Academic Editor

PLOS ONE

Journal Requirements:

Additional Editor Comments (if provided):

Reviewers' comments:

Reviewer's Responses to Questions

**Comments to the Author**

1. Is the manuscript technically sound, and do the data support the conclusions?

Reviewer #1: Yes

Reviewer #2: Yes

2. Has the statistical analysis been performed appropriately and rigorously? 

Reviewer #1: Yes

Reviewer #2: Yes

3. Have the authors made all data underlying the findings in their manuscript fully available?

Reviewer #1: Yes

Reviewer #2: Yes

4. Is the manuscript presented in an intelligible fashion and written in standard English?

Reviewer #1: Yes

Reviewer #2: Yes

5. Review Comments to the Author

Reviewer #1: Please note that due to time constraints，I have not read the MS in full, only the introduction, features of the analysis, and the data and method (briefly).

Intro: Clearly written, no major objections. This type of work has not been completely done before to my awareness.

Features of the analysis: The importance for unreported cases of COVID-19 infections were elaborated in detail. And the author used data from Google Trends to measure the unreported cases appropriately.

Data and method: To my awareness, the data processing and analysis methods are appropriate.

Reviewer #2: I have two main concerns about this paper:

The first is the lack of a theoretical analysis. Can the author theoretically explain whether big data from search engines can really reflect the coronavirus infection rate , how do you demonstrate this with theoretical analysis in the introduction? People's searches on Google are likely to be related to "outbreaks news" rather than "number of cases."

The second concern is: what is the significance of this research, this should be briefly explained in the abstract.

Some smaller points:

1.Can the author make a specific review on the research methods, to prove that the appproach is really innovative?

2.words in paragraph 3 are too much, Can paragraph 3 be merged with paragraph 2, and More emphasis should be placed on methodological innovation.

3.I think the author can delete paragraph 4.

4.In the conclusion part, the author has mentioned the noise factors, but in terms of common sense, these might not the noise factor s and they might probably are the main factors. I think the author need to give the literature basis of this definition, or try to change a word.

6. PLOS authors have the option to publish the peer review history of their article (what does this mean?). If published, this will include your full peer review and any attached files.

Reviewer #1: No

Reviewer #2: No

---

## [Author Response · Author response to Decision Letter 0]

16 Sep 2022

Dear Dr. Ricky Chee Jiun Chia,

Thank you for your feedback on my manuscript and the opportunity to revise my paper “Shed-ding light on dark figures: Steps towards a methodology for estimating actual numbers of COVID-19 infections in Germany based on Google Trends”. The suggestions from the reviewers have been immensely helpful.

Following the two reviews I revised the paper and enhanced the theoretical foundation, the abstract regarding significance of the research and improved some smaller points. The detailed changes made are explained in the next paragraphs. Comments by the reviewers are written in standard font, my responses are in italics and changes to the manuscript are in blue.

The comments and critique have been highly appreciated. Especially the question whether big data based on Google Trends can be a meaningful basis for the research question of my paper was a concern for me in the beginning, too. After doing several statistical analyses and consult-ing literature, I came to the conclusion that this data can indeed serve as a reliable indicator for such questions. The comments helped refine the significance of my work, which is now even better expressed through the changes made.

Finally, I included additional information in my Methods section regarding the datasets and its terms and conditions to comply with the PLOS ONE submission guidelines.

I hope the revised manuscript will better suit PLOS ONE, but I am always open to consider fur-ther revisions. I thank you for your continued interest in my research.

Sincerely,

Christina Maaß

Reviewer Comments, Author Responses and Manuscript Changes

Reviewer #1:

Please note that due to time constraints，I have not read the MS in full, only the introduction, features of the analysis, and the data and method (briefly).

Intro: Clearly written, no major objections. This type of work has not been completely done be-fore to my awareness.

Features of the analysis: The importance for unreported cases of COVID-19 infections were elab-orated in detail. And the author used data from Google Trends to measure the unreported cases appropriately.

Data and method: To my awareness, the data processing and analysis methods are appropriate.

My response:

Thanks to Reviewer #1 for his/her helpful comments. As I read the valuable comments, it made me sensible to add some further details to emphasize the innovation of the paper further.

Reviewer #2:

I have two main concerns about this paper:

Comment 1: The first is the lack of a theoretical analysis. Can the author theoretically explain whether big data from search engines can really reflect the coronavirus infection rate, how do you demonstrate this with theoretical analysis in the introduction? People's searches on Google are likely to be related to "outbreaks news" rather than "number of cases."

My response:

Thanks to Reviewer #2 for his/her helpful comments. While I generally agree that outbreak news can have an influence on search volume, we expect that while phases of outbreak news and stand-ard news alternate, the fear of people to become sick does not weaken over a longer period. We thus believe the utility from Google searches to be higher for individuals with symptoms then for purely interested persons. I have added explanations regarding the theoretical foundations of the empirical analysis, which is based on standard microeconomic theory, to the new manuscript to explain and to emphasize the usefulness of Google search volume to track the number of cases. I thereby refer to the paper written by Lampos et al. (2021) which explicitly show that Google searches are related to the “number of cases” and not only to “outbreak news”. (see lines 79-88 in new manuscript)

Changed text of the manuscript:

The theoretical basis for this econometric paper is standard microeconomic theory, i.e. individ-uals maximizing their utility with the possibility to obtain disease-related information anony-mously. We expect this utility to be higher for individuals with symptoms then for purely inter-ested persons. By estimating the volume of COVID-19-related online searches with autoregres-sive models with and without information about disease-related news articles, [11] show that Google searches are rather representing trends in infection levels than being mainly driven by outbreak news. Although the search volume data surely imply some error in our analysis, it ap-pears that Google Trends data is a viable and highly insightful option to improve standard sta-tistics (that include errors on their own) in order to deal with the complicated task of monitor-ing infection trends (see [11–14]).

Comment 2: The second concern is: what is the significance of this research, this should be briefly explained in the abstract.

My response:

Thank you very much for this advice. I have added information to the abstract that puts more emphasis on the significance of my research. I believe the significance can now be grasped much more easily. The abstract now reads as follows (last sentence added):

Changed text of the manuscript:

In order to shed light on unmeasurable real-world phenomena, we investigate exemplarily the actual number of COVID-19 infections in Germany based on big data. The true occurrence of infections is not visible, since not every infected person is tested. This paper demonstrates that coronavirus-related search queries issued on Google can depict true infection levels appropri-ately. We find significant correlation between search volume and national as well as federal COVID-19 cases as reported by RKI. Additionally, we discover indications that the queries are indeed causal for infection levels. Finally, this approach can replicate varying dark figures throughout different periods of the pandemic and enables early insights into the true spread of future virus outbreaks. This is of high relevance for society in order to assess and understand the current situation during virus outbreaks and for decision-makers to take adequate and jus-tifiable health measures.

Comment 3: Some smaller points:

1.Can the author make a specific review on the research methods, to prove that the approach is really innovative?

My response:

Thank you for this comment. I have added some detail to the introduction that explicitly states the innovation in the paper. The approach is innovative in that it applies the concept of big data analysis to the case of Germany and COVID-19 and approximates the true number of infections (including unreported cases). Neither has the COVID-19 pandemic in Germany been investigated based on Google Trends data, nor has this data been used to investigate the dark figure of COVID-19 infections in general. (see lines 30-34 in new manuscript)

Changed text of the manuscript:

Our approach is novel in that it applies big data analysis, in particular Google Trends search volume analysis, to the specific setting of COVID-19 in Germany and approximates the true number of virus infections (including the dark figure) based on this search volume. To the best of our knowledge, no other study has used this data source to investigate unreported cases of COVID-19.

Comment 4: 2.words in paragraph 3 are too much, Can paragraph 3 be merged with paragraph 2, and More emphasis should be placed on methodological innovation.

My response:

Yes. I have substantially shortened paragraph 3 and merged it with paragraph 2. The first part of paragraph 2 has been merged with paragraph 1. Furthermore, I have explicitly stated the meth-odological innovation of the paper in the new paragraph 2.

Comment 5: 3.I think the author can delete paragraph 4.

My response:

Thank you for the valuable comments on the structure of the first paragraphs. I have considerably cut the number of words in paragraph 4. However, I kept the most relevant information to demon-strate the value of my method for research on dark figures. The first three paragraphs of the man-uscript now read as follows.

Changed text of the manuscript:

There are many economic phenomena that cannot be measured, such as the true extent of il-legal migration, drug trafficking, tax evasion or the spread of diseases. Past research has tackled these dark figures by quantitatively estimating the size of a shadow economy (see e.g. [1]). There, the relationships between the shadow economy and its causes and indicators are calcu-lated with a so-called Multiple Indicators Multiple Causes model and defined with an index of the same name [1]. These proxies were helpful for a better comprehension of hidden markets. Yet with the availability of new data sources that come with the data economy, new opportu-nities are emerging. High-frequency data from social media, search engines, digitized news and information, allows the application of new techniques to shed light on so far unmeasurable activities.

This is where this paper comes in. We develop a methodological approach to investigate a dom-inant topic from the beginning of the 2020s – the coronavirus pandemic, and in particular the dark figure of coronavirus infections in Germany based on information obtained from Google Trends. Our approach is novel in that it applies big data analysis, in particular Google Trends search volume analysis, to the specific setting of COVID-19 in Germany and approximates the true number of virus infections (including the dark figure) based on this search volume. To the best of our knowledge, no other study has used this data source to investigate unreported cases of COVID-19. Since the outbreak of severe acute respiratory syndrome coronavirus type 2 (SARS-CoV-2) (a beta coronavirus identified in early 2020 as the causative agent of COVID-19 [2]) that causes the highly contagious coronavirus disease 2019 (COVID-19) in late 2019, governments and individuals worldwide are attempting to assess the severity of the coronavirus pandemic. For this purpose, they must rely on the number of new confirmed COVID-19 cases as published by health departments. Considering this data more profoundly it appears that the statistic tools used are outdated and problematic. Since not everyone infected with SARS-CoV-2 is automati-cally tested and potentially registered, the true number of infections has to be much higher than official statistics. Additionally, people that are tested do not constitute a random sample that could be extrapolated to the whole population, because the probability to be tested is bi-ased towards symptomatic individuals, so that asymptomatic individuals are underrepresented [3]. Given these difficulties, there is great interest in new methodological approaches to obtain reliable estimates of the actual number of infections; it should be borne in mind that most re-alistic infection quantities are required to understand the dynamics of the pandemic as health measures taken need to be adequate and justifiable.

As a benchmark to get a good picture of the actual number of infections in a country or region, a random sample would be helpful, which, unfortunately, has not been collected. What comes closest are mass tests. However, the positivity rate in such settings appears rather low with 1 % in Slovakia [4] and between 0.1 % and 0.5 % in Austria [5], indicating, for example, about 14 % more infections in Lower Austria than reported when extrapolating this result to the total pop-ulation there. These findings suggest a distorted selection process. Most likely, only individuals with certain characteristics (e.g. asymptomatic and willing to receive a result, or working part-time and thus, at the same time having a lower risk of an infection) took the opportunity to test.

Comment 6: 4.In the conclusion part, the author has mentioned the noise factors, but in terms of common sense, these might not the noise factor s and they might probably are the main factors. I think the author need to give the literature basis of this definition, or try to change a word.

My response:

Yes, I agree with you. Indeed, I used the term noise in a very econometric sense and mean by it factors that can be highly relevant, but that cannot be measured and thus enter the error term. I have made this statement much clearer and explained in more detail how I consider these factors. The first paragraph in the discussion section now reads as follows. (see lines 428-441 in new man-uscript)

Changed text of the manuscript:

We are fully aware that there is some noise (in an econometric sense) in the data set such as some people searching for symptoms or keywords related to the coronavirus may not be in-fected. Furthermore, not everyone uses Google, but other search engines (although this share is negligible), and some people, especially elderly, do not use search engines at all, but are still partly represented by younger people searching on their behalf. Overall, we expect this back-ground noise to be rather constant and not to bias our estimates as long as there are no large and significant differences between the age cohorts with respect to both infections and search activity. This does not mean that these factors do not contain valuable information, but that this variation cannot be explained and enters the error term. The achievement of this paper is to infer information from a big data source even though many relevant factors are unknown. Especially, since we do not look at absolute but relative search volume, we believe we get valu-able insights into the infection situation. Furthermore, the keywords relevant for COVID-19 in-cidence may change over time. This is partially visible in the volume plots. Possible remedies would include keyword selection based on machine learning

---

## [Editor Report · Decision Letter 1]

10 Oct 2022

Shedding light on dark figures: Steps towards a methodology for estimating actual numbers of COVID-19 infections in Germany based on Google Trends

PONE-D-22-02453R1

Dear Dr. Christina Heike Maaß,

We’re pleased to inform you that your manuscript has been judged scientifically suitable for publication and will be formally accepted for publication once it meets all outstanding technical requirements.

Kind regards,

Ricky Chee Jiun Chia

Academic Editor

PLOS ONE
---

## [Editor Report · Acceptance letter]

13 Oct 2022

PONE-D-22-02453R1 

Shedding light on dark figures: Steps towards a methodology for estimating actual numbers of COVID-19 infections in Germany based on Google Trends 

Dear Dr. Maaß:

I'm pleased to inform you that your manuscript has been deemed suitable for publication in PLOS ONE. Congratulations! Your manuscript is now with our production department. 

Kind regards, 

on behalf of

Dr. Ricky Chee Jiun Chia 

Academic Editor

PLOS ONE